# UNDERSTANDING THE LIMITATIONS OF CONDITIONAL GENERATIVE MODELS

**Ethan Fetaya**[*]    **Jörn-Henrik Jacobsen**[*]    **Will Grathwohl**    **Richard Zemel**
Vector Institute and University of Toronto
{ethanf, jjacobs,wgrathwohl, zemel}@cs.toronto.edu

## ABSTRACT

Class-conditional generative models hold promise to overcome the shortcomings of their discriminative counterparts. They are a natural choice to solve discriminative tasks in a robust manner as they jointly optimize for predictive performance and accurate modeling of the input distribution. In this work, we investigate robust classification with likelihood-based generative models from a theoretical and practical perspective to investigate if they can deliver on their promises. Our analysis focuses on a spectrum of robustness properties: (1) Detection of worst-case outliers in the form of adversarial examples; (2) Detection of average-case outliers in the form of ambiguous inputs and (3) Detection of incorrectly labeled in-distribution inputs.

Our theoretical result reveals that it is impossible to guarantee detectability of adversarially-perturbed inputs even for near-optimal generative classifiers. Experimentally, we find that while we are able to train robust models for MNIST, robustness completely breaks down on CIFAR10. We relate this failure to various undesirable model properties that can be traced to the maximum likelihood training objective. Despite being a common choice in the literature, our results indicate that likelihood-based conditional generative models may are surprisingly ineffective for robust classification.

## 1 INTRODUCTION

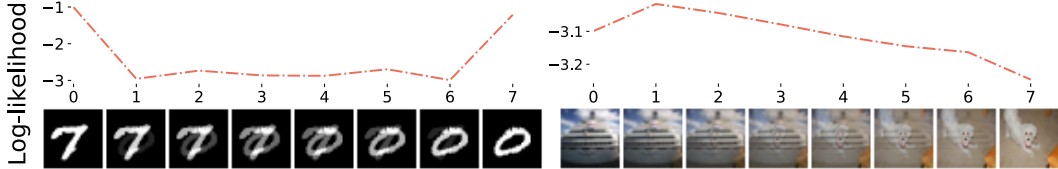

Figure 1: Linear interpolations of inputs and respective outputs of a conditional generative model between two MNIST and CIFAR10 images from different classes. X-axis is interpolation steps and Y-axis negative log-likelihood in bits/dim *(higher is more likely under model)*. MNIST interpolated images are far less likely than real images, whereas for CIFAR10 the opposite is observed, leading to high confidence classification of ambiguous out-of-distribution images.

Conditional generative models have recently shown promise to overcome many limitations of their discriminative counterparts. They have been shown to be robust against adversarial attacks (Schott et al., 2019; Ghosh et al., 2019; Song et al., 2018; Li et al., 2018; Frosst et al., 2018), to enable robust classification in the presence of outliers (Nalisnick et al., 2019b) and to achieve promising results in semi-supervised learning (Kingma et al., 2014; Salimans et al., 2016). Motivated by these success stories, we study the properties of conditional generative models in more detail.

Unlike discriminative models, which can ignore class-irrelevant information, conditional generative models cannot discard any information in the input, potentially making it harder to fool them. Further,

---

[*]equal contribution

jointly modeling the input and target distribution should make it easy to detect out-of-distribution inputs. These traits lend hope to the belief that good class-conditional generative models can overcome important problems faced by discriminative models.

In this work, we analyze conditional generative models by assessing them on a spectrum of robustness tasks. (1) Detection of worst-case outliers in the form of adversarial examples; (2) Detection of average-case outliers in the form of ambiguous inputs and (3) Detection of incorrectly labeled in-distribution inputs. If a generative classifier is able to perform well on all of these, it will naturally be robust to noisy, ambiguous or adversarially perturbed inputs.

Outlier detection in the above settings is substantially different from general out-of-distribution (OOD) detection, where the goal is to use unconditional generative models to detect any OOD input. For the general case, likelihood has been shown to be a poor detector of OOD samples. In fact, often higher likelihood is assigned to OOD data than to the training data itself (Nalisnick et al., 2019a). However, class-conditional likelihood necessarily needs to decrease towards the decision-boundary for the classifier to work well. Thus, if the class-conditional generative model has high accuracy, rejection of outliers from the wrong class via likelihood may be possible.

Our contributions are:

**Provable Robustness**  We answer: *Can we theoretically guarantee that a strong conditional generative model can robustly detect adversarially attacked inputs?* In section 2 we show that even a near-perfect conditional generative model cannot be guaranteed to reject adversarially perturbed inputs with high probability.

**Assessing the Likelihood Objective**  We discuss the basis to empirically analyze robustness in practice. We identify several fundamental issues with the maximum likelihood objective typically used to train conditional generative models and discuss whether it is appropriate for detecting out-of-distribution inputs.

**Understanding Conflicting Results**  We explore various properties of our trained conditional generative models and how they relate to fact that the model is robust on MNIST but not on CIFAR10. We further propose a new dataset where we combine MNIST images with CIFAR background, making the generative task as hard as CIFAR while keeping the discriminative task as easy as MNIST, and investigate how it affects robustness.

## 2  CONFIDENT MISTAKES CANNOT BE RULED OUT

The most challenging task in robust classification is accurately classifying or detecting adversarial attacks; inputs which have been maliciously perturbed to fool the classifier. In this section we discuss the possibility of guaranteeing robustness to adversarial attacks via conditional generative models.

**Detectability of Adversarial Examples**  In the adversarial spheres work (Gilmer et al., 2018) the authors showed that a model can be fooled without changing the ground-truth probability of the attacked datapoint. This was claimed to show that adversarial examples can lie on the data manifold and therefore cannot be detected. While (Gilmer et al., 2018) is an important work for understanding adversarial attacks, it has several limitations with regard to conditional generative models. First, just because the attack does not change the ground-truth likelihood, this does not mean the model can not detect the attack. Since the adversary needs to move the input to a location where the model is incorrect, the question arises: *what kind of mistake will the model make?* If the model assigns low likelihood to the correct class without increasing the likelihood of the other classes then the adversarial attack will be detected, as the joint likelihood over all classes moves below the threshold of typical inputs. Second, on the adversarial spheres dataset (Gilmer et al., 2018) the class supports do not overlap. If we were to train a model of the joint density $p_\theta(x, y)$ (which does not have 100% classification accuracy) then the KL divergence $KL\left(p(x, y)||p_\theta(x, y)\right)$, where $p(x, y)$ is the data density, is infinite due to division by zero (note that $KL\left(p_\theta(x, y)||p(x, y)\right)$ is what is minimized with maximum likelihood). This poses the question, whether small $KL\left(p(x, y)||p_\theta(x, y)\right)$ or small Shannon-Jensen divergence is sufficient to guarantee robustness. In the following, we show that this condition is insufficient.

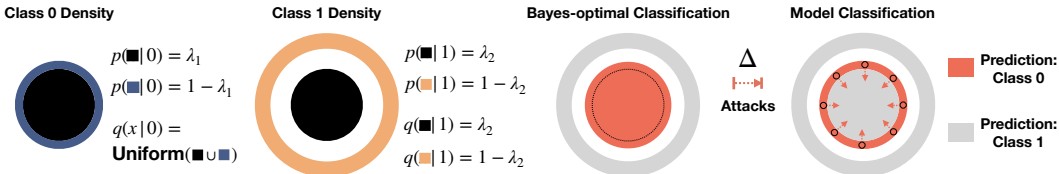

Figure 2: Counter example construction. Shown on the left are the two class data densities, on the right the Bayes-optimal classifier for this problem (assuming $\lambda_1 > \lambda_2$) and the model we consider. Despite being almost optimal, the model can be fooled with undetectable adversarial examples (red arrows). Detailed description in section 2.

**Why no Robustness Guarantee can be Given** The intuition why conditional generative models should be robust is as follows: If we have a robust discriminative model then the set of confident mistakes, i.e. where the adversarial attacks must reside, has low *probability* but might be large in volume. For a robust conditional generative model, the set of undetectable adversarial attacks, i.e. high-density high-confidence mistakes, has to be small in volume. Since the adversary has to be $\Delta$ close to this small volume set, the $\Delta$ area around this small volume set should still be small. This is where the idea breaks down due to the curse of dimensionality. Expanding a set by a small radius can lead to a much larger one even with smoothness assumptions. Based on this insight we build an analytic counter-example for which we can prove that even if

$$KL\left(q||p\right) < \epsilon \quad KL\left(p||q\right) < \epsilon \tag{1}$$

where $p = p(x, y)$ is the data distribution, and $q = q(x, y)$ is the model, we can with probability $\approx 0.5$ take a correctly classified input sampled from $p$, and perturb it by at most $\Delta$ to create an adversarial example that is classified incorrectly and is not detectable.

We note that the probability in every ball with radius $\Delta$ can be made as small as desired, excluding degenerate cases. We also assume that the Bayes optimal classifier is confident and is not affected by the attack, i.e. *we do not change the underlying class but wrongfully flip the decision of the classifier*.

The counter-example goes as follows: Let $U(a, b)$ be the density of a uniform distribution on an annulus in dimension $d$, $\{x \in \mathbb{R}^d : a \leq ||x|| \leq b\}$ then the data conditional distribution is

$$\begin{aligned} p(x|0) =& \quad \lambda_1 U(0,1) + (1 - \lambda_1)U(1, 1 + \Delta) \qquad 0 \leq \lambda_1 \leq 1 \\ p(x|1) =& \quad \lambda_2 U(0,1) + (1 - \lambda_2)U(2, 3) \qquad 0 \leq \lambda_2 \leq 1 \end{aligned} \tag{2}$$

with $p(y = 0) = p(y = 1) = 1/2$. Both classes are a mixture of two distributions, uniform on the unit sphere and uniform on an annulus, as shown in Fig. 2. The model distribution is the following:

$$\begin{aligned} q(x|0) =& \qquad U(0, 1 + \Delta) \\ q(x|1) =& \quad \lambda_2 U(0, 1) + (1 - \lambda_2)U(2, 3) \end{aligned} \tag{3}$$

i.e. for $y = 1$ the model is perfect, while for $y = 0$ we replace the mixture with uniform distribution over the whole domain. If $\lambda_1 \gg \lambda_2$ then points in the sphere with radius 1 should be classified as class $y = 0$ with high likelihood. If $\lambda_2 >> \frac{1}{(1+\Delta)^d}$ then the model classifies points in the unit sphere incorrectly with high likelihood. Finally if $1 >> \lambda_1$ then almost half the data points will fall in the annulus between 1 and $1 + \Delta$ and can be adversarially attacked with distance lesser or equal to $\Delta$ by moving them into the unit sphere as seen in Fig. 2. We also note that these attacks cannot be detected as the model likelihood only increases. In high dimensions, almost all the volume of a sphere is in the outer shell, and this can be used to show that in high enough dimensions we can get the condition in Eq. 1 for any value of $\epsilon$ and $\Delta$ (and also the confidence of the mistakes $\delta$). The detailed proof is in the supplementary material.

This counter-example shows that even under very strong conditions, a good conditional generative model can be attacked. Therefore no theoretical guarantees can be given in the general case for these models. Our construction, however, does not depend on the learning model but on the data geometry. This raises interesting questions concerning the source of the susceptibility to attacks: Is it the model or an inherent issue with the data?

## 3 THE MAXIMUM LIKELIHOOD OBJECTIVE

### 3.1 THE DIFFICULTY IN TRAINING CONDITIONAL GENERATIVE MODELS

Most recent publications on likelihood-based generative models primarily focus on quantitative results of unconditonal density estimation (van den Oord et al., 2016; Kingma & Dhariwal, 2018; Salimans et al., 2017b; Kingma et al., 2016; Papamakarios et al., 2017). For conditional density estimation, either only qualitative samples are shown (Kingma & Dhariwal, 2018), or it is reported that conditional density estimation does not lead to better likelihoods than unconditional density estimation. In fact, it has been reported that conditional density estimation can lead to slightly worse data likelihoods (Papamakarios et al., 2017; Salimans et al., 2017b), which is surprising at first, as extra bits of important information are provided to the model.

**Explaining Likelihood Behaviour** One way to understand this seemingly contradictory relationship is to consider the objective we use to train our models. When we train a generative model with maximum likelihood (either exactly or through a lower bound) we are minimizing the empirical approximation of $\mathbb{E}_{x,y\sim P}\left[-\log(P_\theta(x,y))\right]$ which is equivalent to minimizing $KL(P(x,y)||P_\theta(x,y))$. Consider now an image $x$ with a discrete label $y$, which we are trying to model using $P_\theta(x,y)$. The negative log-likelihood (NLL) objective is:

$$\mathbb{E}_{(x,y)\sim P}[-\log(P_\theta(x,y))] = \quad \mathbb{E}_{x\sim P}[-\log(P_\theta(x))]$$
$$+ \quad \mathbb{E}_{x\sim P}[\mathbb{E}_y[-\log(P_\theta(y|x))|x]] \tag{4}$$

If we model $P_\theta(y|x)$ with a uniform distribution over classes, then the second term has a value of $\log(C)$ where $C$ is the number of classes. This value is negligible compared to the first term $\mathbb{E}_{x\sim P}[-\log(P_\theta(x))]$ and therefore the "penalty" for completely ignoring class information is negligible. So it is not surprising that models with strong generative abilities can have limited discriminative power. What makes matters even worse is that the penalty for confident mis-classification can be unbounded. This may also explain why the conditional ELBO is comparable to the unconditional ELBO (Papamakarios et al., 2017). Another way this can be seen is by thinking of the likelihood as the best lossless compression. When trying to encode an image, the benefit of the label is at most $\log(C)$ bits which is small compared to the whole image. While these few bits are important for users, from a likelihood perspective the difference between the correct $p(y|x)$ and a uniform distribution is negligible. This means that when naively training a class-conditional generative model by minimizing $\mathbb{E}_{(x,y)\sim P}[-\log(P_\theta(x|y))]$, typically discriminative performance as a classifier is very poor.

### 3.2 OUTLIER DETECTION

Another issue arises when models trained with maximum likelihood are used to detect outliers. The main issue is that maximum likelihood, which is equivalent to minimizing $KL(P(x,y)||P_\theta(x,y))$, is known to have a "mode-covering" behavior. It has been shown recently in (Nalisnick et al., 2019a) that generative models, trained using maximum likelihood, can be quite poor at detecting out-of-distribution example. In fact it has been shown that these models can give higher likelihood values, on average, to datasets different from the test dataset that corresponds to the training data. Intuitivily one can still hope that a high accuracy conditional generative model would recognize an input conditioned on the wrong class as an outlier, as it was successfully trained to separate these classes. In section 4.2 we show this is not the case in practice.

While (Nalisnick et al., 2019a) focuses its analysis into dataset variance, we propose this is an inherit issue with the likelihood objective. If it is correct then the way conditional generative models are trained is at odds with their desired behaviour. If this is the case, then useful conditional generative model will require a fundamentally different approach.

## 4 EXPERIMENTS

We now present a set of experiments designed to test the robustness of conditional generative models. All experiments were performed with a flow model where the likelihood can be computed in closed form as the probability of the latent space embedding (the prior) and a Jacobian correction term; see Sec A.1 for a detailed explanation. Given that we can compute $p(x,y)$ for each class, we can easily compute $p(y|x)$ and classify accordingly. Besides allowing closed-form likelihood computation, the

flexibility in choosing the prior distribution was important to conduct various experiments. In our work we used a version of the GLOW model; details of the models and training is in the supplementary material sec. B. We note that the results are not unique to flow models, and we verified that similar phenomenon can be seen when training with the PixelCNN++ autoregressive model (Salimans et al., 2017a) in sec. E.

## 4.1 TRAINING CONDITIONAL GENERATIVE MODELS

Here we investigate the ability to train a conditional generative model with good likelihood and accuracy simultaneously. Usually in flow models the prior distribution in latent space $z$ is Gaussian. For classification we used aclass-conditional mixture of 10 Gaussians $p(z|y) = \mathcal{N}(\mu_y, \sigma_y^2)$ We compare three settings: 1) A class-conditional mixture of 10 Gaussians as the prior (Base). 2) A class-conditional mixture of 10 Gaussians trained with an additional classification loss term (Reweighted). 3) Our proposed conditional split prior (Split) described in sec. A.4 in the supplementary material. Results can be found in table 1.

| **MNIST** | Base | Reweight | Split | **CIFAR10** | Base | Reweight | Split |
|---|---|---|---|---|---|---|---|
| *% Acc* | 96.9 | 99.0 | 99.3 | *% Acc* | 56.8 | 83.2 | 84.0 |
| *bits/dim*[1] | 0.95 | 1.10 | 1.00 | *bits/dim* | 3.47 | 3.54 | 3.53 |

Table 1: Comparison between different models.

As we can see, especially on CIFAR10, pushing up the accuracy to values that are still far from state-of-the-art already results in non-negligible deterioration to the likelihood values. This exemplifies how obtaining strong classification accuracy without harming likelihood estimation is still a challenging problem. We note that while the difference between the split prior and re-weighted version is not huge, the split prior achieves better NLL and better accuracy in both experiments. We experimented with various other methods to improve training with limited success, see sec. C in the supplementary material for furture information.

## 4.2 NEGLIGIBLE IMPACT OF CLASS MEMBERSHIP ON LIKELIHOOD

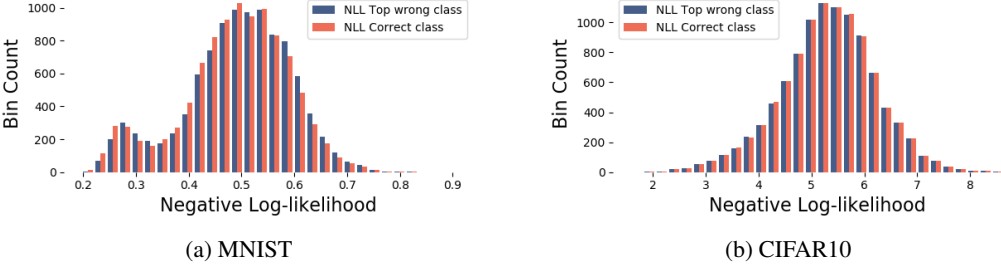

(a) MNIST

(b) CIFAR10

Figure 3: NLL for images conditioned on the correct class vs the highest probability wrong class.

Next we show that even conditional generative models which are strong classifiers do not see images with the corrupted labels as outliers. To understand this phenomenon we first note that if we want the correct class to have a probability of at least $1 - \delta$ then it is enough for the corresponding logit to be larger than all the others by $\log(C) + \log\left(\frac{1-\delta}{\delta}\right)$ where $C$ is the number of classes. For $C = 10$ and $\delta = 1e - 5$ this is about 6, which is negligible relative to the likelihood of the image, which is in the scale of thousands. This means that even for a strong conditional generative model which confidently predicts the correct label, the pair $\{x_i, y_w \neq y_i\}$ (where $w$ is the leading incorrect class) cannot be detected as an outlier according to the joint distribution, as the gap $\log(p(x_i|y_i)) - \log(p(x_i|y_w))$ is much smaller than the variation in likelihood values. In Fig. 3 we show this by plotting the histograms of the likelihood conditioned both on the correct class and on the most likely wrong class over the test set. In other words, in order for $\log(p(x_i|y_w))$ to be considered an outlier the prediction needs to be extremely confident, much more than we expect it to be, considering test classification error.

---

[1] As we model zero padded datasets as described in section A.4, these numbers are not exactly comparable with the literature.

### 4.3 ADVERSARIAL ATTACKS AS WORST CASE ANALYSIS

We first evaluate the ability of conditional generative models to detect standard attacks, and then try to detect attacks designed to fool the detector (likelihood function). We evaluate both the gradient based Carlini-Wagner $L_2$ attack (CW-$L_2$) (Carlini & Wagner, 2017b) and the gradient free boundary attack (Brendel et al., 2018). Results are shown in table 2 on the left. It is interesting to observe the disparity between the CW-$L_2$ attack, which is easily detectable, and the boundary attack which is much harder to detect.

| **Attacking** | **Classification** | | **Classification and Detection** | |
|---|---|---|---|---|
| **MNIST** | Reweight | Split | Reweight | Split |
| $CW - L_2$ | 0% (100%) | 1% (100%) | 17% (100%) | 14% (100%) |
| Boundary attack | 43% (82%) | 36% (80%) | 0% (0%) | 0% (0%) |
| **CIFAR10** | | | | |
| $CW - L_2$ | 0% (97%) | 0% (0%) | 6% (99%) | 3% (100%) |
| Boundary attack | 67% (100%) | 72% (100%) | 100% (100%) | 100% (100%) |

Table 2: Comparison of attack detection. Percentage of successful and undetected attacks within $L_2$-distance of $\epsilon = 1.5$ for MNIST and $\epsilon = 33/255$ for CIFAR10 for proposed models. Number in parentheses is percentage of attacks that successfully fool the classifier, both detected and undetected.

Next we modify our attacks to try to fool the detector as well. With the CW-$L_2$ attack we follow the modification suggested in (Carlini & Wagner, 2017a) and add an extra loss term $\ell_{det}(x') = \max\{0, -\log(p(x')) - T\}$ where $T$ is the detection threshold. For the boundary attack we turn the $C$-way classification into a $C + 1$-way classification by adding another class which is "non-image" and classify any image above the detection threshold as such. We then use a targeted attack to try to fool the network to classify the image into a specific original class. This simple modification to the boundary attack will typically fail because it cannot initialize. The standard attack starts from a random image and all random images are easily detected as "non-image" and therefore do not have the right target class. To address this we start from a randomly chosen image from the target class, ensuring the original image is detected as a real image from the desired class.

From table 2 (right side) we can see that even after the modification CW-$L_2$ still struggles to fool the detector. The boundary attack, however, succeeds completely on CIFAR10 and fails completely on MNIST, even when it managed to sometimes fool the detector without directly trying. We hypothesize that this is because the area between two images of separate classes, where the boundary attack needs to pass through, is correctly detected as out of distribution only for MNIST and not CIFAR10. We explore this further below.

### 4.4 AMBIGUOUS INPUTS AS AVERAGE CASE ANALYSIS

To understand why the learned networks are easily attacked on CIFAR but not on MNIST with the modified boundary attack, we explore the probability density of interpolations between two real images. This is inspired by the fact that the boundary attack proceeds along the line between the attacked image and the initial image. The minimum we would expect from a decent generative model is to detect the intermediate middle images as "non-image" with low likelihood. If this was the case and each class was a disconnected high likelihood region, the boundary attack would have a difficult time when starting from a different class image.

Given images $x_0$ and $x_1$ from separate classes $y_0$ and $y_1$ and for $\alpha \in [0, 1]$ we generate an intermediate image $x_\alpha = \alpha \cdot x_1 + (1 - \alpha)x_0$, and run the model on various $\alpha$ values to see the model prediction along the line. For endpoints we sample real images that are classified correctly and are above the detection threshold used previously. See Fig. 1 for interpolation examples from MNIST and CIFAR.

In figure 4 (a) we see the average results for MNIST for 1487 randomly selected pairs. As expected, the likelihood goes down as $\alpha$ moves away from the real images $x_0$ and $x_1$. We also see the probability of both classes drop rapidly as the network predictions become less confident on the intermediate images. Sampling 100 $\alpha$ values uniformly in the range $[0, 1]$ we can also investigate how many of the

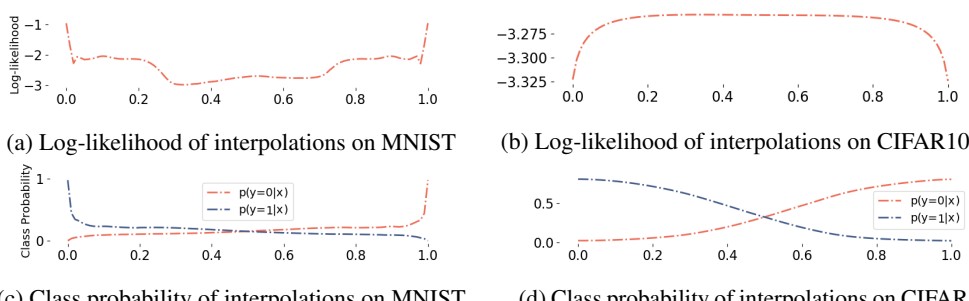

(a) Log-likelihood of interpolations on MNIST  (b) Log-likelihood of interpolations on CIFAR10

(c) Class probability of interpolations on MNIST  (d) Class probability of interpolations on CIFAR10

Figure 4: Average Log likelihoods and class probabilities for interpolations between data points from different classes, x-axis is interpolation coefficient $\alpha$. The MNIST model behaves as desired and robustly detects interpolated images. The CIFAR10 model, however, fails strikingly and interpolatd images are consistently more likely than true data under the model.

interpolations all stay above the detection threshold, i.e. all intermediate images are considered real by the model, and find that this happens only in 0.5% of the cases.

On CIFAR images, using 1179 pairs, we get a very different picture (see fig. 4 (b)). Not only does the intermediate likelihood not drop down, it is even higher on average than on the real images albeit to a small degree. In classification we also see a very smooth transition between classes, unlike the sharp drop in the MNIST experiment. Lastly, 100% of the interpolated images lay above the detection threshold and none are detected as a "non-image" (for reference the detection threshold has 78.6% recall on real CIFAR10 test images). This shows that even with good likelihood and reasonable accuracy, the model still "mashes" the classes together, as one can move from one Gaussian to another without passing through low likelihood regions in-between. It also clarifies why the boundary attack is so successful on CIFAR but fails completely on MNIST. We note that the basic attack on MNIST is allowed to pass through these low density areas which is why it sometimes succeeds.

### 4.5 CLASS-UNRELATED ENTROPY IS TO BLAME

In this section, we show that the difference in performance between CIFAR10 and MNIST can largely be attributed to how the entropy in the datasets is distributed, i.e how much the uncertainty in the data distribution is reduced after conditioning on the class label. For MNIST digits, a large source of uncertainty in pixel-space comes from the class label. Given the class, most pixels can be predicted accurately by simply taking the mean of the training set in each class. This is exactly why a linear classifier performs well on MNIST. Conversely on CIFAR10, after conditioning on the class label there still exists considerable uncertainty. Given the class is "cat," there still exists many complicated sources of uncertainty such as where the cat is and how it is posed. In this dataset, a much larger fraction of the uncertainty is not accounted for after conditioning on the label. This is not a function of the domain or the dimensionality of the dataset, it is a function of the dataset itself.

To empirically verify this, we have designed a dataset which replicates the challenges of CIFAR10 and places them onto a problem of the same *discriminative* difficulty as MNIST. To achieve this, we simply replaced the black backgrounds of MNIST images with randomly sampled (downsampled and greyscaled) images from CIFAR10. In this dataset, which we call background-MNIST (BG-MNIST), the classification problem is identically predictable from the same set of pixels

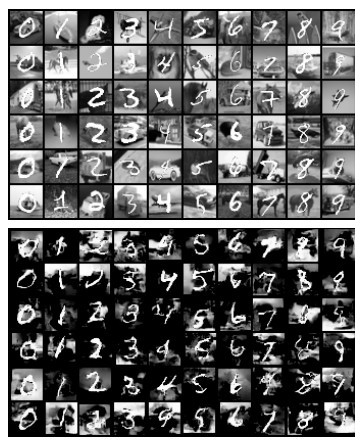

Figure 5: Top: Samples from the BG-MNIST-0 dataset. Bottom: Samples from conditional generative model trained on the dataset. Note how the model has learnd to capture digit identity.

as in standard MNIST but modeling the data density is much
more challenging.

To further control the entropy in a fine-grained manner, we convolve the background with a Gaussian
blur filter with various bandwidths to remove varying degrees of high frequency information. With
high blur, the task begins to resemble standard MNIST and conditional generative models should
perform as they do on MNIST. With low and no blur we expect them to behave as they do on
CIFAR10.

Table 3 summarizes the performance of conditional generative models on BG-MNIST. We train
models with a "Reweighted" discriminative objective as in Section A. The reweighting allows them
to perform well as classifiers but the likelihood of their generative component falls to below CIFAR10
levels. More strikingly, now when we interpolate between datapoints we observe behavior identical
to our CIFAR10 models. This can be seen in Figure 6. Thus, we have created a dataset with the
discriminative difficulty of MNIST and the generative difficulty of CIFAR10.

|  | MNIST | BG-MNIST-5 | BG-MNIST-1 | BG-MNIST-0 | CIFAR10 |
|---|---|---|---|---|---|
| *% Acc* | 99 | 99 | 99 | 98 | 84 |
| *bits/dim* | 1.10 | 1.67 | 3.30 | 4.58 | 3.53 |

Table 3: Conditional generative models trained on BG-MNIST. BG-MNIST-$X$ indicates the bandwith
of blur applied to CIFAR10 backgrounds.

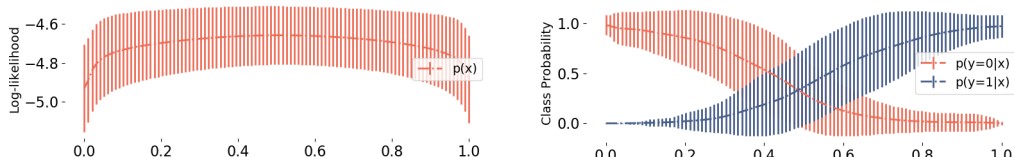

(a) Log-likelihood of interpolations on BG-MNIST-0   (b) Class probability of interpolations on BG-MNIST-0

Figure 6: Average log-likelihoods and class probabilities for interpolations between BG-MNIST-0
datapoints. While classification is on par with MNIST models, the likelihood exhibits the same
failures as CIFAR10 models.

## 5   RELATED WORK

Despite state of the art performance in many tasks, deep neural networks have been shown to be
fragile where small image transformations, (Azulay & Weiss, 2018) or background object transplant
(Rosenfeld et al., 2018) can greatly change predictions. In the more challenging case of adversarial
pertubations, deep neural networks are known to be vulnerable to adversarial attacks (Akhtar & Mian,
2018), and while many attempts have been made to train robust models or detect malicious attacks,
significant progress towards truly robust models has been made only on MNIST (Schott et al., 2019;
Madry et al., 2017). Even CIFAR10 remains far from being solved from a standpoint of adversarial
robustness.

One common belief is that adversarial attacks succeed by moving the data points off the data manifold,
and therefore can possibly be detected by a generative model which should assign them low likelihood
values. Although this view has been challenged in (Gilmer et al., 2018), we now discuss how their
setting needs to be extended to fully study robustness guarantees of conditional generative models.

Recent work (Song et al., 2018; Frosst et al., 2018; Li et al., 2018) showed that a generative model
can detect and defend adversarial attacks. However, there is a caveat when evaluating detectability of
adversarial attacks: the attacker needs to be able to attack the detection algorithm as well. Not doing
so has been shown to lead to drastically false robustness claims (Carlini & Wagner, 2017a). In (Li
et al., 2018) the authors report difficulties training a high accuracy conditional generative model on
CIFAR10, and resort to evaluation on a 2-class classification problem derived from CIFAR10. While

they do show robustness similar to our Carlini-Wagner results, they do not apply the boundary attack which we found to break our models on CIFAR10. This highlights the need to utilize a diverse set of attacks. In (Schott et al., 2019) a generative model was used not just for adversarial detection but also robust classification on MNIST, leading to state-of-the-art robust classification accuracy. The method was only shown to work on MNIST, and is very slow at inference time. However, overall it provides an existence proof that conditional generative models can be very robust in practice. In (Ghosh et al., 2019) the authors also use generative models for detection and classification but only show results with the relatively weak FGSM attack, and on simple datasets. As we see in Fig. 1 and discuss in section 4, generative models trained on MNIST can display very different behavior than similar models trained on more challenging data like CIFAR10. This shows how success on MNIST may often not translate to success on other datasets.

## 6 CONCLUSION

In this work we explored limitations, both in theory and practice, of using conditional generative models to detect adversarial attacks. Most practical issues arise due to likelihood, the standard objective and evaluation metric for generative models by which probabilities can be computed. We conclude that likelihood-based density modeling and robust classification may fundamentally be at odds with one another as important aspects of the problem are not captured by this training and evaluation metric. This has wide-reaching implications for applications like out-of-distribution detection, adversarial robustness and generalization as well as semi-supervised learning with these models.

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

# A  TRAINING CONDITIONAL GENERATIVE MODELS

## A.1  LIKELIHOOD-BASED GENERATIVE MODELS AS GENERATIVE CLASSIFIERS

We present a brief overview of flow-based deep generative models, conditional generative models, and their applications to adversarial example detection.

Flow based generative models Rezende & Mohamed (2015); Dinh et al. (2015; 2017); Kingma & Dhariwal (2018) compute exact densities for complex distributions using the change of variable formula. They achieve strong empirical results Kingma & Dhariwal (2018) and the closed form likelihood makes them easier to analyze than the closely related VAE Kingma et al. (2014). The main idea behind flow-based generative models is to model the data distribution using a series of bijective mappings $z_N = f(x) = f_N \circ f_{N-1}... \circ f_1(x)$ where $z_N$ has a known simple distribution, e.g. Gaussian, and all $f_i$ are parametric functions for which the determinant of the Jacobian can be computed efficiently. Using the change of variable formula we have $\log(p(x)) = \log(p(z_N)) + \sum_{i=1}^{N} \log(|\det(J_i(z_i))|)$ where $J_i$ is the Jacobian of $f_i$ and $z_{i-1} = f_{i-1}... \circ f_1(x)$.

The standard way to parameterize such functions $f_i$ is by splitting the input $z_{i-1}$ into two $z_{i-1} = (z_{i-1}^1, z_{i-1}^2)$ and chose

$$f_i\left(\begin{bmatrix} z_{i-1}^1 \\ z_{i-1}^2 \end{bmatrix}\right) = \begin{bmatrix} z_{i-1}^1 \\ s(z_{i-1}^1) \odot z_{i-1}^2 + t(z_{i-1}^1) \end{bmatrix} \tag{5}$$

which is invertible as long as $s(z_{i-1}^1)_j = s_j \neq 0$ and we have $\log(|\det(J_i(z_{i-1}))|) = \sum_j \log(|s_j|)$. For images the splitting is normally done in the channel dimension. These models are then trained by maximizing the empirical log likelihood (MLE).

A straightforward way to turn this generative model into a conditional generative model is to make $p(z_N)$ a Gaussian mixture model (GMM) with one Gaussian per class, i.e. $p(z_N|y) = \mathcal{N}(\mu_y, \Sigma_y)$. Assuming $p(y)$ is known, then maximizing $\log(p(x, y))$ is equivalent to maximizing $\log(p(x|y)) = \log(p(z_N|y)) + \sum_{i=1}^{N} \log(|\det(J_i(z_i))|)$. At inference time, one can classify by simply using Bayes rule. Note that directly optimizing $\log(p(x|y))$ results in poor classification accuracy as discussed in section 3. This issue was also addressed in the recent hybrid model work Nalisnick et al. (2019a).

We will now describe some of the various approaches we investigated in order to train the best possible flow-based conditional generative models, to achieve a better trade-off betwen classification accuracy and data-likelihood as compared to commonly-used approaches. We also discuss some failed approaches in the appendix.

## A.2  REWEIGHTING

The most basic approach, which has been used before in various works, is to reweight the discriminative part in eq. (4). While this can produce good accuracy, it can have an unfavorable trade-off with the NLL where good accuracy comes with severely sub-optimal NLL. This tradeoff has also been shown in Nalisnick et al. (2019a) where they train a somewhat similar model but classify with a generalized linear model instead of a Gaussian mixture model.

## A.3  ARCHITECTURE CHANGE

Padding channels has been shown to increase accuracy in invertible networks Jacobsen et al. (2018); Behrmann et al. (2019). This helps ameliorate a basic limitation in bijective mappings (see Eq. (5)), by allowing to increase the number of channels as a pre-processing step. Unlike the discriminative i-RevNet, we cannot just pad zeros as that would not be a continuous density. Instead we pad channels with uniform(0,1) random noise. In effect we do not model MNIST and CIFAR10 as is typically done in the literature, but rather the zero-padded version of those. While the ground-truth likelihoods for the padded and un-padded datapoints are the same due to independence of the uniform noise and unit density of the noise, this is not guaranteed to be captured by the model, making likelihoods very similar but not exactly comparable with the literature. This is not an issue for us, as we only compare models on the padded datasets.

### A.4 SPLIT PRIOR

One reason MLE is bad at capturing the label is because a small number of dimensions have a small effect on the NLL. Fortunately, we can use this property to our advantage. As the contribution of the conditional class information is negligible for the data log likelihood, we choose to model it in its distinct subspace, as proposed by Jacobsen et al. (2019). Thus, we partition the hidden dimensions $z = (z_s, z_n)$ and only try to enforce the low-dimensional $z_s$ to be the logits. This has two advantages: 1) we do not enforce class-conditional dimensions to be factorial; and 2) we can explicitly up-weight the loss on this subspace and treat it as standard logits of a discriminative model. A similar approach is also used by semi-supervised VAEs Kingma et al. (2014). This lets us jointly optimize the data log-likelihood alongside a classification objective without requiring most of the dimensions to be discriminative. Using the factorization $p(z_s, z_n|y) = p(z_s|y) \cdot p(z_n|z_s, y)$ we model $p(z_s|y)$ as Gaussian with class conditional mean $e_i = (0, ..., 0, 1, ...0)$ and covariance matrix scaled by a constant. The distribution $p(z_n|z_s, y)$ is modeled as a Gaussian where the mean and variances are a function of $y$ and $z_n$.

## B IMPLEMENTATION DETAILS

We pad MNIST with zeros so both datasets are 32x32 and subtract 0.5 from both datasets to have a [-0.5,0.5] range. For data augmentation we do pytorch's random crop with a padding of 4 and 'edge' padding mode, and random horizontal flip for CIFAR10 only.

The model is based on GLOW with 4 levels, affine coupling layers, 1x1 convolution permutations and actnorm in a multi-scale architecture. We choose 128 channels and 12 blocks per level for MNIST and 256 channels and 16 blocks for CIFAR10. In both MNIST and CIFAR10 experiments we double the number of channels with uniforrm(0,1) noise which we scale down to the range $[0, 2/256]$ (taking it into account in the Jacobian term). One major difference is that we do the squeeze operation at the end of each level instead of the beginning, which is what allows us to use 4 levels. This is possible because with the added channels the number of channels is even and the standard splitting is possible before the squeeze operation.

The models are optimized using Adam, for 150 epochs. The initial learning rate is $1e - 3$, decayed by a factor of 10 every 60 epochs. For the reweighted optimization the objective is

$$loss = -\log(p(x|y))/D - \log(p(y|x)) \qquad (6)$$

where D is the data dimension (3x32x32 for CIFAR, 1x32x32 for MNIST).

For adversarial detection we use a threshold of 1.4 for MNIST (100% of test data are below the threshold) and 4. for CIFAR10 (78.6% of test images are below the threshold).

## C NEGATIVE RESULTS

In this work explored many ideas in order to achieve better tradeoff between accuracy with little or no impact.

### C.1 ROBUST PRIORS

Since the Gaussian prior is very sensitive to outliers, one idea was that confident miss-classifications carry a strong penalty which might result in "messing" all the classes together. A solution would be to replace the Gaussian with a more robust prior, e.g. Laplace or Cauchy. Another idea we explored is a mixture of Gaussian and Laplace or Cauchy using the same location parameter. In our experiments we did not see any significant difference from the Gaussian prior.

### C.2 LABEL SMOOTHING

Another approach to try to address the same issue is a version of label smoothing. In this new model the Gaussian clusters are a latent variable that is equal to the real label with probability $1 - \epsilon$

and uniform on the other labels with probability $\epsilon$. Using this will bound the error for confident miss-classification as long as the data is close to one of the Gaussian centers.

### C.3 FLOW-GAN

As we claimed the main issue is with the MLE objective, it seems like a better objective is to optimize $KL\left(p(x,y)||p_\theta(x,y)\right)$ or the Jensen-Shannon divergence as this KL term is highly penalized for miss-classification. It is also more natural when considering robustness against adversarial attacks. Optimizing this directly is hard, but generative adversarial networks (GANs) Goodfellow et al. (2014) in theory should also optimize this objective. Simply training a GAN would not work as we are interested in the likelihood value for adversarial detection and GANs only let you sample and does not give you any information regarding an input image.

Since flow algorithms are bijective, we could combine the two objective as was done in the flow-GAN paper Grover et al. (2018). We trained this approach with various conditional-GAN alternatives and found it very hard to train. GANs are know to be unstable to train, and combining them with the unstable flow generator is problematic.

### D  ANALYTICAL COUNTER EXAMPLE:

Assume $p(y=1) = p(y=0) = q(y=1) = q(y=0) = 1/2$ and

$$p(x|0) = \lambda_1 U(0,1) + (1-\lambda_1)U(1,1+\Delta) \tag{7}$$
$$p(x|1) = \lambda_2 U(0,1) + (1-\lambda_2)U(2,3) \tag{8}$$
$$q(x|0) = U(0,1+\Delta) \tag{9}$$
$$q(x|1) = \lambda_2 U(0,1) + (1-\lambda_2)U(2,3) \tag{10}$$

where $U(a,b)$ is the uniform distribution on the annulus $R^d(a,b) = \{x \in \mathbb{R}^d : a \leq ||x|| \leq b\}$ in dimension $d$.

**Lemma 1.** *For $||x|| < 1$ we have*

$$p(0|x) = \frac{\lambda_1}{\lambda_1 + \lambda_2} \tag{11}$$

$$q(0|x) = \frac{1}{1 + \lambda_2(1+\Delta)^d} \tag{12}$$

*Proof.* The $U(a,b)$ density (when it isn't zero) is $\frac{1}{C_d(b^d-a^d)}$ where $c_d$ is the volume of the $d$-dimensional unit ball. The proof follows by a simple use of Bayes rule. $\qquad\square$

so by having $\lambda_1 >> \lambda_2 >> \frac{1}{(1+\Delta)^d}$ we can have the model switch wrongfully predictions from $y=0$ to $y=1$ when we move $x$ from the annulus $R^d(1,1+\Delta)$ to $R^d(0,1)$

**Lemma 2.** *If $\lambda_1 > \frac{1}{(1+\Delta)^d}$ and $\lambda_1 < 1 - e^{-\epsilon}$ then $KL(q(x,y)||P(x,y)) \leq \epsilon$*

*Proof.* Using the chain rule for KL divergence, $\mathrm{KL}(P(x,y)||Q(x,y)) = \mathrm{KL}(P(y)||Q(y)) + \mathbb{E}_y[\mathrm{KL}(P(x|y)||Q(x|y))]$ we get that $\mathrm{KL}(q(x,y)||P(x,y)) = \mathrm{KL}(q(x|y=0)||P(x|y=0))$. We now have

$$\mathrm{KL}(q(x|y=0)||P(x|y=0)) = \int_{R^d(0,1)} \frac{1}{C_d(1+\Delta)^d} \log\left(\frac{\frac{1}{C_d(1+\Delta)^d}}{\frac{\lambda_1}{C_d}}\right) \tag{13}$$

$$+ \int_{R^d(1,1+\Delta)} \frac{1}{C_d(1+\Delta)^d} \log\left(\frac{\frac{1}{C_d(1+\Delta)^d}}{\frac{1-\lambda_1}{C_d((1+\Delta)^d-1)}}\right) = \frac{-\log(\lambda_1(1+\Delta)^d)}{(1+\Delta)^d} \tag{14}$$

$$+ \frac{(1+\Delta)^d - 1}{(1+\Delta)^d} \log\left(\frac{(1+\Delta)^d - 1}{(1-\lambda_1)(1+\Delta)^d}\right) \leq \log\left(\frac{1}{1-\lambda_1}\right) < \epsilon \tag{15}$$

$$\square$$

**Lemma 3.** *If* $1 > \lambda_1 > \frac{1}{(1+\Delta)^d}$ *and* $\lambda_1 < \frac{\epsilon}{d\log(1+\Delta)}$ *then* $KL(P(x,y)||q(x,y)) \le \epsilon$

*Proof.* Again using the KL chain rule we have

$$\text{KL}(P(x|y=0)||q(x|y=0)) = \lambda_1 \int_{R^d(0,1)} \frac{1}{C_d} \log\left(\frac{\frac{\lambda_1}{C_d}}{\frac{1}{C_d(1+\Delta)^d}}\right) \tag{16}$$

$$\int_{R^d(1,1+\Delta)} \frac{(1-\lambda_1)}{C_d((1+\Delta)^d - 1)} \log\left(\frac{\frac{(1-\lambda_1)}{C_d((1+\Delta)^d-1)}}{\frac{1}{C_d(1+\Delta)^d}}\right) \le \lambda_1 d\log(1+\Delta) < \epsilon \tag{17}$$

$$\square$$

**Proposition 1.** *For all* $(\epsilon, \delta, \Delta)$ *there is a distribution* $p$ *and an approximation* $q$ *in dimension* $d = \tilde{\mathcal{O}}\left(\frac{\log(\frac{\delta}{1+\delta})+\log(\frac{1}{\epsilon})}{\log(1+\Delta)}\right)$ *such that*

$$KL(q(x,y)||p(x,y)) < \epsilon, \quad KL(p(x,y)||q(x,y)) < \epsilon \tag{18}$$

*but with probability greater then* $1/3$ *over samples* $x \sim p$ *there is an adversarial example* $\bar{x}$ *satisfying:*

1. *$y_q(x) = y_p(x)$ with $p(y_p(x)|x)$ and $q(y_q(x)|x)$ greater or equal to $1 - \delta$. The original point is classifier correctly and confidently.*

2. *$y_q(x) \neq y(\bar{x})$, $y_q(\bar{x}) = y(\bar{x})$. We change the prediction without changing the ground-truth label.*

3. *$q(y_q(\bar{x})|\bar{x}) < \delta$, $p(y_p(\bar{x})|\bar{x}) > 1 - \delta$. The classifier is confident in its wrong prediction.*

4. *$||x - \bar{x}|| < \Delta$. We make a small change to the inputs.*

5. *The density $q(\bar{x})$ is greater or equal to the median density, making the attack undetectable by observing $q(x)$.*

6. *For $\Delta < 1$ the probability in any radius ball can be made as small as desired.*

7. *The total variation of the distribution can be made as small as desired.*

The last two conditions exclude degenerate trivial counter-exmaples, one where the whole distribution support is in a $\Delta$ radius ball and $\Delta$ does indeed represent a small pertubation. The other condition excludes "pathological" distributions ,e.g. misclassification on a dense zero measure set like the rationals.

*Proof.* In order to satisfy conditions 1-5, using previous lemmas, it is enough that

1. $\frac{\lambda_1}{\lambda_1 + \lambda_2} \ge 1 - \delta$

2. $\frac{1}{1 + \lambda_2(1+\Delta)^d} \le \delta$

3. $\lambda_1 \le 1 - e^{-\epsilon}$

4. $\lambda_1 > \frac{1}{(1+\Delta)^d}$

5. $\lambda_1 < \frac{\epsilon}{d\log(1+\Delta)}$

By setting $\lambda_2 = \frac{\delta}{1-\delta}\lambda_1$ we can easily satisfy condition 1. It is not hard to see that condition 2 is equivalent to $\lambda_1 \ge \left(\frac{1-\delta}{\delta}\right)^2 \frac{1}{(1+\Delta)^d}$ which superseeds condition 4 when $\delta < 1/2$. Condition 3 can be

satisfied with $\lambda_1 < \epsilon/2$ by using $1 - x \geq e^{-2x}$ for $x < 1/2$.

This boils down to ensuring $d$ is large enough so that there is a valid $\lambda_1$ such as

$$\left(\frac{1-\delta}{\delta}\right)^2 \frac{1}{(1+\Delta)^d} < \lambda_1 < \frac{\epsilon}{d\log(1+\Delta)} \tag{19}$$

Which is true for large enough $d$ as the l.h.s decays exponentially while the r.h.s linearly.

Condition 6 is trivial as the radius of the support is fixed so as long as $\Delta < 1$ the probability in any $\Delta$ radius ball decays exponentially. Regarding total variation, we note that from the divergence theorem this can be bounded by a term that depends on the surface area of shperes with fixed radius which decreases to zero as $d$ goes to infinity.

$\square$

## E   PIXELCNN++

We trained a conditional PixelCNN++ where instead of predicting each new pixel using a mixture of 10 components, we use one mixture component per class. Using reweighting we train using the following objective $-log(p(x|y))/dim + \alpha \cdot -log(p(y|x))$. As one can see from table 4, standard trainig, i.e. $\alpha = 0$, results in very poor accuracy, while reweighting the classification score results in much better accuracy but worse NLL.

| $\alpha$ | acc (%) | bits/dim |
|------|---------|----------|
| 0 | 25.48 | 3.05 |
| 1000 | 85.78 | 3.34 |

Table 4: Accuracy and NLL for pixelCNN++ on CIFAR10

