# OpenReview forum: "Understanding the Limitations of Conditional Generative Models"
_ICLR.cc/2020/Conference — Accept (Poster)_

### Official Review · AnonReviewer2 · 2019-10-17
**Official Blind Review #2**

**Rating:** 6

**Review:**

Post rebuttal:

Thank you for your response. I appreciate the authors add an experiment on BG-MNIST, which shows the intermediate trend of MNIST and CIFAR-10.

As the authors mentioned, the reweighting scheme could be a simple yet effective way to address the problem of current likelihood-based models. While there is room for improvement to further develop the method, the current version of the paper would be a good contribution to the community.

Hence, I raise my score from 3 to 6.

----------------------------------------

Summary:
This paper investigates some limitations of the conditional generative models (or generative classifiers). First, the authors present a counter-example that a good generative classifier fails to detect adversarial attacks. Second, the authors claim that the marginal and conditional terms of the likelihood objective are the source of the problem. Finally, the authors demonstrate some experiments on adversarial attacks, out-of-distribution (OOD) samples, and noisy labels.

Pros:
- While generative classifiers are believed to be more robust than the discriminative counterparts [1], the authors present a counter-example that it may not be true.
- The authors investigate the marginal and conditional terms of the likelihood objective and demonstrate empirical results that the model fails to capture the outliers.

Cons:

1. The imbalance issue of the likelihood objective is not surprising.

As the data x is far complex than the class y, it is expectable that the penalty from modeling p(x) is larger than the penalty from classifying p(y|x). As mentioned in Table 1 and Appendix A.2, balancing two terms indeed improves the classification performance. However, to meet the high standard of ICLR, the authors should propose an alternative or modification of the likelihood which resolves the existing limitations. For example, [2] decomposes the semantic and background parts to improve the OOD detection using likelihood models.

2. The experiments are not extensively studied.

The authors conduct experiments on two datasets: MNIST and CIFAR-10. The authors may present more results on other datasets (e.g., SVHN or CIFAR-100) and convince if their findings are consistent. Also, some observations seem to be an inheritance of the datasets, e.g., Figure 4 is natural since MNIST has disjoint support and CIFAR-10 has a continuous one.

Minor comments:
- On page 8, ',' should be moved after (Azulay & Weiss, 2018).
- On page 8, (Schott et al.) should be changed to '\citet' format.
- PixelCNN++ is doubly cited.


[1] Li et al. Are Generative Classifiers More Robust to Adversarial Attacks? ICML 2019.
[2] Ren et al. Likelihood Ratios for Out-of-Distribution Detection. NeurIPS 2019.


**Experience Assessment:**

I have read many papers in this area.

**Review Assessment: Checking Correctness Of Derivations And Theory:**

I assessed the sensibility of the derivations and theory.

**Review Assessment: Checking Correctness Of Experiments:**

I assessed the sensibility of the experiments.

**Review Assessment: Thoroughness In Paper Reading:**

I read the paper at least twice and used my best judgement in assessing the paper.

---

> ### Author Response · Authors · 2019-11-15
> **Reply Blind Review #2**
>
> We thank the reviewer for the detailed comments, we have updated the paper accordingly and believe your comments were very helpful to improve the manuscript.
>
> Regarding specific points:
>
> Q: The imbalance issue of the likelihood objective is not surprising
> A: While the imbalance is indeed simple when you think about it, we found that many researchers underestimate how bad “vanilla” conditional generative models are at classification, making it an important point to make. We do explore alternatives, e.g. split prior (others are in the supplementary material) and find empirically that more sophisticated approaches don't outperform the reweighting approach. Moreover, we do not consider this one of the major contributions of this work but a simple yet important observation we think should be discussed.
>
> Q: Adding more datasets
> A: While more experiments are always better we do believe that experiments on MNIST and CIFAR10 are standard practice in the adversarial attacks community (e.g. [1]). Furthermore, while the first robust models on MNIST have recently been suggested, CIFAR10 is still very far from being solved. Our paper aims to shed light on this discrepancy from a conditional generative modeling perspective. We believe this is important to understand, as this model class has shown great promise and the state-of-the-art for robust classification on MNIST is a conditional generative model as well.
>
> To add more insight to the difference between MNIST and CIFAR10 we do however add another dataset that mixes them, BG-MNIST. This dataset allowed us to study the observed discrepancy in a more controlled setup.
>
> Q: Figure 4 interpolations are natural and an inheritance of the dataset
> A: We agree that the properties shown in figure 4 are a consequence of the dataset. In fact, this was the point we tried to make. However, we disagree that figure 4 is natural. The mixture pictures are outliers to humans (see for example fig.1) and should have low density with a good generative model. It may well be possible that disjoint support plays a role here as you mentioned. However, it is not obvious that the class-conditional densities are not disjoint in the CIFAR10 case, it is hard to find images in CIFAR10 that are truly ambiguous in terms of their ground-truth label, where it is easy in MNIST (e.g. 1s and 7s).
>
> Furthermore, we have introduced the BG-MNIST task where we purposefully make MNIST more like CIFAR10 by adding CIFAR images in the background of MNIST digits. Surprisingly, doing so does not cause the model to behave like it did on the pure CIFAR10 task despite having even worse test set likelihoods in the most extreme case than the CIFAR10 model. So it is not straightforward to explain what the difference between MNIST and CIFAR10 is. In fact, the difference does not only seem to be related to entropy or disjoint support but also strongly coupled with the nature of the discriminative task.
>
> To make this more clear we have added new plots to the paper that show how different BG-MNIST models behave when doing linear interpolation between samples from different classes. These plots show interesting behaviour: we can observe that log-likelihood is lower for interpolated images, just like on MNIST, even though CIFAR10 dominates the image content. However, the fraction of interpolated images that can easily be detected by thresholding log-likelihoods depends on the complexity of the CIFAR10 background, which we control with the blur kernel.
>
> Q: Minor comments
> A: Thank you for the minor comments, we will fix the mistakes
>
> [1]  A. Madry, A. Makelov, L. Schmidt, D. Tsipras, and A. Vladu, “Towards deep learning models resistant to adversarial attacks,” in International Conference on Learning Representations, 2018.

---

### Official Review · AnonReviewer3 · 2019-10-23
**Official Blind Review #3**

**Rating:** 6

**Review:**

Update: I thank the authors' for their response, and have read the other reviews.

This paper demonstrates some theoretical and practical limitations on the use of likelihood based generative models for detecting adversarial examples. They construct a simple counterexample showing that there are adversarial examples for an arbitrarily accurate model (as measured by KL) that are not detectable by diminished likelihood of the model (as the dimension increases). Extending the work of Gilmer et al, this proves that there can be no general robustness guarantee for conditional generative models (Bayes classifiers). They provide compelling empirical evidence that while conditional normalizing flows trained on MNIST can be effective in detecting and defending adversarial attacks, these models trained on CIFAR10 are not. Surprisingly, it is shown that linear interpolations between images of different classes yield higher likelihoods for the CIFAR10 models and that class has little impact on model likelihoods. This goes some way in explaining why the detection is not effective on CIFAR, but questions still remain.

The paper makes fairly modest claims, but does a good job at demonstrating them and shedding some light on the issue. The experiments are thorough and fit into a growing body of evidence that the likelihoods of normalizing flows and other image based likelihood models may not be that informative or well calibrated, where past work has focused on out-of-distribution detection. My only major complaint with the paper is that it is not clear to what extent the theoretical and practical problems are related. As mentioned in the paper, the counterexample construction depends on the geometry of the data rather than the learning model. It could be that for both the MNIST and CIFAR10 datasets, the geometry is such that robustness garauntees are possible, and that the discrepancy in detection and interpolation arises because the normalizing flow has modeled the MNIST distribution much better than the CIFAR10 distribution. In this case we might hope that using conditional likelihood models for adversarial detection can be made effective, but that effort needs to be placed into improving the modeling capability. It's not obvious how to probe this distinction, but it would be good if this was given some thought in the paper. Also it would be good to see the attack detection numbers on BG-MNIST.


Comments:
Difficulty in training conditional generative models:
I believe in the two papers you cite the models do not use the label as input, but rather there is a separate model for each class? The overfitting is likely why the models had slightly lower conditional likelihood. As an aside, there are a couple of other examples of conditional normalizing flow models on images that use a mixture of Gaussians in the latent space [1], [2].

eq. 4: In the paper it is said that the second term in eq 4 is at most log(C), because the uniform distribution would have this value and that therefore this is negligibly small in comparison to the other term. Why exactly is this the case, couldn't the data entropy term be smaller in principle even if it's larger in practice? Or is the argument that the data entropy term scales with the dimensionality, but the label term does not leading to an imbalance? This could use some clarification.

A3: What is meant by ‘While the ground-truth likelihoods for the padded and un-padded datapoints are the same due to independence of the uniform noise and unit density of the noise’ in the appendix section A3? Wouldn't the ground truth negative log likelihoods would increase by the entropy of the uniform noise? Also, then in the bits per dimension calculations is the dimension the number of unpadded dimensions or the padded dimensions?


[1] Izmailov, Pavel, et al. "Semi-Supervised Learning with Normalizing Flows." Workshop on Invertible Neural Nets and Normalizing Flows, International Conference on Machine Learning. 2019.
[2] Atanov, Andrei, et al. "Semi-Conditional Normalizing Flows for Semi-Supervised Learning." Workshop on Invertible Neural Nets and Normalizing Flows, International Conference on Machine Learning. 2019.

**Experience Assessment:**

I have published one or two papers in this area.

**Review Assessment: Checking Correctness Of Derivations And Theory:**

I assessed the sensibility of the derivations and theory.

**Review Assessment: Checking Correctness Of Experiments:**

I carefully checked the experiments.

**Review Assessment: Thoroughness In Paper Reading:**

I read the paper at least twice and used my best judgement in assessing the paper.

---

> ### Author Response · Authors · 2019-11-15
> **Reply Blind Review #3**
>
> We thank the reviewer for the detailed comments. We have incorporated them into the manuscript.
>
> Regarding specific points:
>
> Q: It is not clear if better models will allow for robustness on CIFAR10
> A: We have also analyzed PixelCNN++, it has significantly higher likelihoods than Glow and shows the same behaviour. We have also added more results on the BG-MNIST task, which includes CIFAR10 in the background. The results show that models with better and worse likelihoods than the pure CIFAR10 model can both have more favourable behaviour in terms of how the likelihoods behave on interpolated images.
>
> This means that simply improving likelihoods is not necessarily going to solve the problem. We have shown that no guarantees can be given and this makes it hard to predict the behaviour in practice. We do believe though that we as humans are robust to epsilon perturbations, so robust classification is possible. However, knowing that no guarantees can be given for these models is an important contribution and makes extensive empirical investigation necessary in practice.
>
> Q: Cited conditional generative models use separate model per class
> A: Both the masked autoregressive flow and pixelCNN++ we cite use a single model for all classes, feeding in the class information as another input for each layer.
>
> Q: Other examples in literature use mixture of Gaussian as well
> A: Thank you for bringing these references to our attention. We do not claim using a mixture of Gaussians as part of the novelty of this paper. It is the most straightforward approach so it is of little surprise it has been done before. We will cite these extra papers.
>
> Q: Is the argument that data entropy scales with dimensionality and labels do not?
> A: Indeed the problem is that the data scales with dimensionality while the classification is fixed. We have made this clearer in the text.
>
> Q: Clarify the padded uniform noise
> A: By padding the data x, with extra dimensions of uniform noise, our data is now (x, u), this defines a distribution p(x, u) = p(x)p(u) and since we are using uniform noise p(u)=1, so p(x, u) = p(x). Now, our generative model is modeling p(x, u), so if the marginal p(u) under our model is not perfectly uniform, then this equality no longer holds. However, this is not a problem for our evaluation since all models compared in our work use the same amount of noise and padding.

---

### Official Review · AnonReviewer1 · 2019-10-29
**Official Blind Review #1**

**Rating:** 6

**Review:**

paper summary:
The authors claim that likelihood based generative models are not as robust to noise as general consensus claims them to be. To prove this authors make use of adversarial, ambiguous and incorrectly labeled in distribution inputs. Authors address issues regarding robustness in near perfect conditional generative models as well as assess the robustness of the likelihood objective.

Pros of the paper:
1) Authors make well motivated arguments about how a near perfect generative model is also susceptible to attacks by providing examples that are adversarial, and have high likelihood and yet are incorrectly labeled.
2) They also demonstrate how class conditional generative models have poor discriminative power.

Cons:
1) The experiments section is written very poorly. This section relies heavily on the supplement making it hard to read due to the constant back and forth between the results and details of the experiments.
2) Experiments seem largely limited. Comparisons on image  data sets such as MNIST and CIFAR10 alone are not convincing enough to establish generalizability of the proposed theory. For example, the hypothesis could completely fail on text based generative models.



**Experience Assessment:**

I do not know much about this area.

**Review Assessment: Checking Correctness Of Derivations And Theory:**

I did not assess the derivations or theory.

**Review Assessment: Checking Correctness Of Experiments:**

I assessed the sensibility of the experiments.

**Review Assessment: Thoroughness In Paper Reading:**

I read the paper at least twice and used my best judgement in assessing the paper.

---

> ### Author Response · Authors · 2019-11-15
> **Reply Blind Review #1**
>
> We thank the reviewer for the effort and remarks. We incorporated the suggested changes and believe that they have significantly improved the manuscript.
>
> Regarding the two “cons”:
>
> Q: Experimental section heavily relies on supplement and is thus hard to read
> A: We have thoroughly revised the experimental section and moved some core material back into the main body of the text, to minimize the overhead caused by cross-references. We initially put part of the experimental details in the supplement to try to stay within the soft 8 pages limit, hoping that the material in the paper is enough for an intuitive understanding of the experiments.
>
> Q: Datasets sufficient to support our claims?
> A: MNIST and CIFAR are standard datasets for adversarial attacks so experimenting on them is aligned with common practice. Furthermore, while the first robust models on MNIST have recently been suggested, CIFAR10 is still very far from being solved. Our paper aims to shed light on this discrepancy from a conditional generative modeling perspective. We believe this is important to understand, as this model class has shown great promise and the state-of-the-art for robust classification on MNIST is a conditional generative model as well. However, we also added our new BG-MNIST dataset that combines MNIST and CIFAR10 and gives another point of view. Regarding text based generative models, as text is discrete the concept of epsilon perturbation isn’t appropriate and it very common in the literature to only study robustness on images. Therefore, we leave this type of data for future work.
>
> To increase the value of the BG-MNIST experiments, we have added some more interpolation results and discussion on them in the manuscript. The results highlight that the difference between CIFAR10 and MNIST is quite subtle in parts, as the surprising interpolation behaviour on CIFAR10 can not just be explained by excessive class-unrelated entropy or poor likelihoods of the model.

---

### Author Response · Authors · 2019-11-15
**Revision Summary**

We thank the reviewers for their time and comments.
Following their insights and suggestions, we have thoroughly revised the draft.

We have:

=>  Improved the experimental section to rely less on the appendix and make it is easier to follow.
=>  Added suggested references and discussion.
=>  Updated and added results to the BG-MNIST section and discussed them. They show how the difference between MNIST and CIFAR10 is difficult to attribute to a single property of these datasets.
=>  Discussed that these new results again highlight how likelihood fails to give us the full picture. A model trained on BG-MNIST, a similarly challenging density modeling task as CIFAR10, can perform considerably worse than a CIFAR10 model in terms of likelihood on in-distribution data, but still behave more favourably in terms of the interpolations through ambiguous inputs. This suggests a complex interplay between difficulty of discriminative and generative parts of the objective.
=>  Clarified other points raised by the reviewers.

---

### Decision · Program_Chairs · 2019-12-19

**Decision:**

Accept (Poster)

**Comment:**

This paper presents theoretical results showing the conditional generative models cannot be robust. The paper also provide counter examples and some empirical evidence showing that the theory is reflected in practice. Some reviewers doubt how much of the theory holds in reality, but still they think that this paper could be a useful for the community. After the rebuttal period, R2 increased their score and it seems that with the current score the paper can be accepted.